# Physiological and Biochemical Responses of Pseudocereals with C3 and C4 Photosynthetic Metabolism in an Environment with Elevated CO_2_

**DOI:** 10.3390/plants13233453

**Published:** 2024-12-09

**Authors:** Bruna Evelyn Paschoal Silva, Stefânia Nunes Pires, Sheila Bigolin Teixeira, Simone Ribeiro Lucho, Natan da Silva Fagundes, Larissa Herter Centeno, Filipe Selau Carlos, Fernanda Reolon de Souza, Luis Antonio de Avila, Sidnei Deuner

**Affiliations:** 1Department of Botany, Biology Institute, Federal University of Pelotas, Pelotas 96010-610, RS, Brazil; brunabiologia89@hotmail.com (B.E.P.S.); stefanianunespires@gmail.com (S.N.P.); sheilabigolin@gmail.com (S.B.T.); simonibelmonte@gmail.com (S.R.L.); natanfagundes@gmail.com (N.d.S.F.); 2Department of Soils, Faculty of Agronomy Eliseu Maciel, Federal University of Pelotas, Pelotas 96010-610, RS, Brazil; larissa.centeno@ufpel.edu.br (L.H.C.); filipeselaucarlos@hotmail.com (F.S.C.); 3Department of Plant and Soil Sciences, Mississippi State University, Mississippi State, MS 39762, USA; fr278@msstate.edu (F.R.d.S.); luis.avila@pss.msstate.edu (L.A.d.A.)

**Keywords:** *Amaranthus* spp., *Chenopodium quinoa* (Willd), climate change, photosynthetic parameters, carbohydrate metabolism

## Abstract

The present work aimed to investigate the effect of increasing CO_2_ concentration on the growth, productivity, grain quality, and biochemical changes in quinoa and amaranth plants. An experiment was conducted in open chambers (OTCs) to evaluate the responses of these species to different levels of CO_2_ {*a*[CO_2_] = 400 ± 50 μmol mol^−1^ CO_2_ for ambient CO_2_ concentration, *e*[CO_2_] = 700 ± 50 μmol mol^−1^ CO_2_ for the elevated CO_2_ concentration}. Growth parameters and photosynthetic pigments reflected changes in gas exchange, saccharolytic enzymes, and carbohydrate metabolism when plants were grown under *e*[CO_2_]. Furthermore, both species maintained most of the parameters related to gas exchange, demonstrating that the antioxidant system was efficient in supporting the primary metabolism of plants under *e*[CO_2_] conditions. Both species were taller and had longer roots and a greater dry weight of roots and shoots when under *e*[CO_2_]. On the other hand, the panicle was shorter under the same situation, indicating that the plants invested energy, nutrients, and all mechanisms in their growth to mitigate stress in expense of yield. This led to a reduction on panicle size and, ultimately, reducing quinoa grain yield. Although *e*[CO_2_] altered the plant’s metabolic parameters for amaranth, the plants managed to maintain their development without affecting grain yield. Protein levels in grains were reduced in both species under *e*[CO_2_] in the average of two harvests. Therefore, for amaranth, the increase in CO_2_ mainly contributes to lowering the protein content of the grains. As for quinoa, its yield performance is also affected, in addition to its protein content. These findings provide new insights into how plants C3 (amaranth) and C4 (quinoa) respond to *e*[CO_2_], significantly increasing photosynthesis and its growth but ultimately reducing yield for quinoa and protein content in both species. This result ultimately underscore the critical need to breed plants that can adapt to *e*[CO_2_] as means to mitigate its negative effects and to ensure sustainable and nutritious crop production in future environmental conditions.

## 1. Introduction

With the advent of innovative techniques, it has become possible to detect even minor climate changes; however, these alterations have occurred over the past 2 million years [1]. In the past, gas releases may have occurred naturally from massive seabed deposits of methane hydrates, emissions from volcanic eruptions, or decay of vegetation associated with asteroid impacts. Nowadays, anthropogenic emissions of greenhouse gases (GHGs) may have similar effects. These emissions have massive effects on the global carbon cycle and are causing significant climate changes [2].

The concentration of GHGs in the Earth’s atmosphere is directly linked to the average global temperature of the Earth, which, at the same time, has been rising steadily since the time of the Industrial Revolution, and the most abundant gas, accounting for two-thirds of GHGs is CO_2_, the most common byproduct of fossil fuel burning [3].

Results of large-scale experiments presented significant variations and showed that an increase in CO_2_ does not necessarily promote plant growth, varying from species to species [4,5]. These changes, considered positive in a short time, mainly promote the improvement of water use efficiency (amount of organic matter produced by the amount of water used) in both C4 and C3 plants.

The impacts of climate change on plant physiology and the increase in global population have raised serious concerns about food security [6]. Moreover, Brazil is a special case in this context because agriculture is a key economic activity, and most people working in agriculture in the country are family farmers. Smallholders help to maintain the diversity of agricultural products and nutrients worldwide, collaborating with food security in this scenario [7].

The most direct and simplest adaptive measure would be to encourage these farmers to conduct research that supports the implementation of different cultures [8]. In addition, smallholders should preferentially cultivate products with higher added value, such as Pseudocereals. The most likely replacement products are derived from protein-rich crops, including pseudocereals. Although these crops have declined in production and consumption over several decades, they have achieved significant recognition in recent years due to their effectiveness and sustainability [9].

According to the American Heritage Dictionary of the English language, the term pseudocereals can be defined as any plant that does not belong to the grass family. On the other hand, these plants produce fruits and seeds that are usually used as flour for bread and other staple foods [10]. Besides that, like cereals, they have starchy, dry seeds and are much more protein-rich. Plants usually included in the non-systematic grouping of pseudocereals are dicotyledonous and belong to different families. Consequently, plants such as quinoa and amaranth are placed in the Chenopodiaceae and Amaranthaceae family, which belongs to the Caryophyllales order, a subclass of Caryophyllidae [11,12].

Given the growing evidence that the research about changes and impacts in climate is leading to a better understanding of the potential pressures on the ability to ensure an adequate food supply for the human population, comprehending how plants respond is essential to precede studies related to these effects on the plant physiology [13].

Evidence that elevated carbon dioxide concentration {*e*[CO_2_]} can cause rapid growth has been proved over the years since the first research in the 1970s and 1980s [14,15]. These alterations happen because photosynthetic carbon gain results in an enhanced carbohydrate metabolism. Consequently, there are alterations in plants' assimilate partitioning and sink strength [16,17].

As an important component of source–sink, the model of Munch postulates that unloading and loading of the conducting tissue are mainly driven by concentration and/or osmotic gradients [18,19]. From another point of view, the source–sink can be considered as the competitive ability of an organ to import photoassimilates [20]. However, studies about how the carbohydrate metabolism of pseudocereals responds to *e*[CO_2_] and what may be the impacts of the alteration of the source–sink function resulting from the effect of the increase in the concentration of atmospheric carbon on the nutrition, metabolism, and productivity of quinoa and amaranth plants are incipient.

Therefore, this study aimed to verify the effects of *e*[CO_2_] in amaranth (*Amaranthus* spp.) and quinoa (*Chenopodium quinoa*, Willd) plants, considering that these species have C3 and C4 metabolism, respectively. Here, it was demonstrated in a 2-year experiment that the relationship of physiological parameters such as growth and photosynthesis are intimately linked with source–sink, sucrose metabolism, and nutrient content influencing the grains’ productivity.

Considered the main greenhouse gas (GHG), atmospheric carbon dioxide (CO_2_) concentration has increased progressively since the Industrial Revolution. The last IPCC report indicated that the National Oceanic and Atmospheric Administration (NOAA) estimated a global [CO_2_] was 421 ppm [2]. In addition to being a “greenhouse” gas, CO_2_ is also the source of carbon for plant photosynthesis and growth, and the ongoing increase in its concentration has been shown to stimulate a wide range of plant species. However, the extent of any stimulation can be related to photosynthetic biochemistry, with C3 plants generally showing a more robust response than C4 plants. Indirect effects, primarily through *e*[CO_2_] closing stomata, can also improve water use efficiency for both C3 and C4 species [4,21].

Understanding the behavior and adaptation of alternative crops to climate change is essential because crop diversity is key to adapting to climate change [7,13]. This is especially true for small family farms [8]. One important option for diversity in small family farms is to grow protein-rich crops, including pseudocereals. These pseudocereals have been recognized in recent years due to their nutritional values [9]. Quinoa and amaranth are important pseudocereals belonging to the Caryophyllales order, a subclass of Caryophyllidae from the Chenopodiaceae and Amaranthaceae families, respectively [12]. They differ in photosynthetic biochemistry and potential response to *e*[CO_2_], with quinoa being a C3 species and amaranth a C4.

Quinoa and amaranth are considered inexpensive and abundant sources of digestive fiber, protein containing methionine and lysine, vitamin C, carotenoids, and minerals. It is also a great source of antioxidant pigments, such as betacyanin, betaxanthin, betalain, amaranthine, and bioactive phytochemicals, including flavonoids and phenolic acids. These bioactive components of natural origin can quench ROS [22]. Humans and animals have extensively consumed these crops in the Andes for millions of years [23] and have been studied recently for their ability to develop in adverse conditions such as high temperature, drought, and salinity [24,25]. Although, at present, there are data regarding how quinoa and amaranth can respond to rising CO_2_ and climate change, there are still fundamental unknowns regarding primary metabolism, including changes in nutritional quality, antioxidant capacity, and associations with photosynthetic and growth parameters. Therefore, this study aimed to assess the effects of future high CO_2_ *e*[CO_2_] on quinoa and amaranth.

## 2. Materials and Methods

### 2.1. Growth Conditions

Seeds of the cultivar BRS Alegria (amaranth) and BRS Piabiru (quinoa) were sown in polystyrene trays on commercial substrate (Plantmax^®^, Cascavel, PR, Brazil). After the second pair of true leaves appeared, the seedlings were transplanted into 8 L polyethylene pots filled with soil, which was previously analyzed for its physical and chemical attributes, amended, and fertilized according to technical recommendations. In the transition from the vegetative to the reproductive stage, leaves were collected for subsequent growth, physiological, and biochemical analyses.

#### 2.1.1. Description of the Experimental Site (OTCs)

The experiments were conducted using open-top chambers (OTC) from the Weed Science Center of the Federal University of Pelotas (Capão do Leão—RS). These chambers are equipped with sensors, an automated CO_2_ concentration control center, coolers responsible for homogenizing the air inside them, and a gas injection and distribution valve system in each chamber. The OTC has a useful area of 4 m^2^ and 2.15 m in height, is coated with a 150-micron thick, transparent polyethylene plastic film, and is equipped with a top-opening reducer to deflect the air and prevent the dilution of the desired concentration of CO_2_ inside the chamber. Carbon dioxide (Messer^®^, Canoas, RS, Brazil) used was 99.9% pure and was supplied through a storage cylinder (capacity of 25 kg CO_2_) coupled to the injection and distribution system of the chambers.

#### 2.1.2. Elevated CO_2_ Treatment

Two experiments were conducted separately in 2019/2020 and 2020/2021. Four OTCs with two different pseudocereals (amaranth and quinoa) were employed in each experimental run. The plants were grown at two levels, 400 ± 50 μmol mol^−1^ of CO_2_ {*ambient* CO_2_ concentration = *a*[CO_2_]} and 700 ± 50 μmol mol^−1^ of CO_2_ {*elevated* CO_2_ concentration, = *e*[CO_2_]} until the experiment was finished (Appendix A). In short, two studies have been conducted with pseudocereal species at different levels of CO_2_ (400 and 700 μmol mol^−1^). The internal temperature of the OTCs within each experimental year was monitored daily using a data logger (HOBO Pro v2), as shown in Appendix A. The mean daily relative humidity (RH) from December 1 to 31 was 62.92% in 2019/2020 and 71.75% in 2020/2021. Concerning RH (minimum/maximum’) was 36.70/80.00%’ in 2019/2020 and 53.50/87.70% in 2020/2021.

### 2.2. Growth Parameters

Shoot (SL) and root length (RL): Measured using a graduated ruler (cm). Shoot (SDM) and root (RDM) dry matter: Obtained by drying the samples in an oven at 65 °C until constant weight (mg plant^−1^). Stem diameter: Measured using a digital caliper (mm). Panicle length (PL): Evaluated using a graduated ruler (cm). Branches per panicle (BPP): Obtained by counting the number of branches per panicle per plant. Leaf area (cm^2^) was estimated using the following methodologies:

Amaranth leaf area (LA): Measured using the equation 2HC/3, as described by Monteiro et al. [26], where 2/3 is the form factor determined for amaranth leaves, while H and C indicate the largest leaf dimensions in the longitudinal and transversal directions. The last pair of expanded leaves were used.

Quinoa leaf area (LA): Measured using the equation F (LxC), where F corresponds to the correction factor 0.6079 described by Benincasa [27] and L and C indicate the largest leaf dimensions in the longitudinal and transversal directions. The last pair of expanded leaves were used.

### 2.3. Physiological and Biochemical Parameters

Photosynthetic pigments: The photosynthetic pigments were quantified according to the methodology proposed by Welburn [28]. For this purpose, one leaf disc was obtained from two young-expanded leaves per experimental unity, sampling ten repetitions per treatment. The leaves were cut into small segments, using 0.01 g of fresh sample inserted into test tubes containing 3.5 mL of dimethyl sulfoxide (DMSO) neutralized with 5% calcium carbonate. Then, the tubes were incubated in a water bath at a temperature of 65 °C for 1 h, protected from light, and then cooled in the dark until reaching room temperature. After, absorbance readings at 480 nm, 649 nm, and 665 nm were taken in a spectrophotometer. Chlorophyll *a*, *b*, total, and carotenoid contents were calculated based on the equations: chlorophyll *a* = (12.47 × A665) − (3.62 × A649); chlorophyll *b* = (25.06 × A649) − (6.5 × A665); carotenoids = (1000 × A480) − (1.29 × chlorophyll a) − (53.78 × chlorophyll b)/220; and the results were expressed in mg g^−1^ FW.

Leaf gas exchange: The leaf gas exchange was determined using an Infrared Gas Analyzer LI 6400 XT (LI-COR Environmental, Lincoln, NE, USA). The evaluation was carried out between 8:30 and 10:00 a.m. The concentration of CO_2_ in the chamber was matched for each treatment (400 and 700 μmol mol^−1^ CO_2_), and the photon flux density was regulated to 1500 µmol of photons m^−2^ s^−1^ with a light source attached to the measuring chamber. Moreover, the leaf temperature ranged from 26 to 27 °C in 2019/2020 and 26 to 29 °C in 2020/2021. Regarding the VPD leaf, the mean was 1.76 and 1.89 in 2019/2020 and 2020/2021, respectively. Net CO_2_ assimilation (*A*), stomatal conductance (*g_s_*), internal concentration of CO_2_ (*Ci*), and transpiration rate (*E*) were measured using the middle third of the youngest expanded leaf. Water use efficiency (*WUE*) was obtained through the *A/E* ratio. Leaf gas exchange measurements were performed only once for each experiment. The analysis was performed in the transition period, i.e., vegetative to flowering.

Total soluble sugars (TSS), starch, sucrose (SUC), and total soluble amino acids (SAA) in leaves: The collected material was standardized, using approximately 250 mg of the middle third of two fully expanded leaves, with four repetitions per treatment. After being weighed, the material was macerated in 8 mL of extracting solution M:C:W (methanol: chloroform: ultra-pure water in the proportion of 12:5:3) and stored in amber flasks for 24 h in the dark. After this period, 2 mL of M:C:W solution was added, and the extract was centrifuged at 2500× *g* for 30 min. After centrifugation, 8 mL of the supernatant was transferred to Falcon tubes, and 2 mL of chloroform and 3 mL of milli-Q water were added. The falcons were centrifuged again for 30 min at 2500× *g* for phase separation. The upper phase was collected and concentrated by evaporation to approximately 50% of the volume at 30 °C to eliminate the excess methanol and chloroform residues present. The extract obtained at the end was later used for quantification of TSS [29], SUC [30]and SAA [31].

After drying at room temperature, the precipitate obtained from the first centrifugation was resuspended in 8 mL of 10% (*w:v*) trichloroacetic acid (TCA). In the above precipitate, 10 mL of 30% perchloric acid was added. After stirring for 30 min, the tubes containing the reaction medium were centrifuged at 2500 rpm for 30 min. Starch was quantified from the collected supernatant [29].

Quantification of TSS was completed using test tubes with screw caps bathed in ice. After adding the extracts diluted in pure water, 1.5 mL of anthrone solution (0.15% in concentrated sulfuric acid) was added to each tube. After 15 min, the tubes were shaken and incubated at 90 °C for 20 min. After that, the tubes were kept in the dark until reaching room temperature. Starch determination was performed in the same way as TSS. At the end of the process, the values obtained were multiplied by the correction factor of 0.9 for conversion into starch contents. The determination of PSA was carried out using the same methodology as AST. Readings were performed in a spectrophotometer at wavelengths of 620 nm for total soluble sugars, starch, water-soluble polysaccharides, and sucrose and 570 nm for total soluble amino acids. For the quantification of sucrose, test tubes with screw caps bathed in ice, extracts were used 100 μL of 30% KOH was transferred to tubes. The tubes were incubated in a water bath for 10 min at 100 °C. After reaching room temperature, 3 mL of anthrone (0.15% in 70% sulfuric acid) were incubated again in a water bath at 40 °C for 15 min.

SAA contents were determined from extracts plus 0.5 mL of 0.2 M citrate buffer pH 5.0, 0.2 mL of 5% ninhydrin reactive in ethylene glycol monomethyl ether, and 1 mL of 2% (*v/v*) KCN in methyl cellosolve (prepared from the 0.01 M KCN solution in pure water). The capped test tubes were incubated in a water bath at 100 °C for 20 min. After 20 min at room temperature, 1.3 mL of 60% ethanol was added.

Determination of sucrose metabolism-related enzyme activity: Leaf samples from the apical portions of the plants (about 0.4 g) were ground to a fine powder in the presence of liquid N_2_. The extraction of the neutral/alkaline invertase (CINV) and the acid invertase enzymes (CWINV and VINV) followed the methodology described by Zeng et al. [32], with minor modifications. In each sample, 1.5 mL of extractor medium containing potassium phosphate buffer (200 mM, pH 7.5), PMSF (1 mM), MgCl_2_ (5 mM), DTT (1 mM), and ascorbic acid (50 mM) was added and then centrifuged at 18,000× *g* for 20 min at 4 °C. The supernatant solution was collected to measure soluble invertase activity (VINV and CINV), and the precipitate was collected to measure insoluble invertase (CWINV). In addition to the reagents used for the soluble invertases, NaCl (1 M) and Tritonne-X-100 (1%) were also added for CWINV. The enzyme extract (500 μL) was added to a 1000 μL assay medium containing 500 μL sodium acetate buffer (pH 4.5 for VINV and CWIN activity and pH 7.5 for CINV activity), 200 mM sucrose, and 5 mM MgCl_2_. The incubation temperature was 37 °C, and 200 μL aliquots were collected after 10 and 40 min to determine enzymatic activity. Enzymatic activity was evaluated by quantifying reducing sugars produced according to the dinitrosalicylic acid (DNS) method described by Miller [33]. All enzyme activities were determined in triplicate and expressed in micromoles of glucose per gram of fresh weight per min (µmol glucose g^−1^ FW min^−1^).

Susy’s activity was determined according to Lowell et al. [34], with some modifications. The enzymatic extract of Susy was prepared using 0.5 g of homogenized samples, 0.05 M HEPES (pH 7.0), 1 mM EDTA (Ethylenediamine tetraacetic acid), 2 mM MgCl_2_ and DTT (dithiothreitol), 0.1 M ascorbic acid and water. The homogenate was centrifuged at 13,000× *g* for 20 min at 4 °C. Then, 100 µL of extract (supernatant) was added to 1900 µL of the medium containing 0.1 M morpholino ethanesulfonic acid (MES) buffer (pH 6.0), 0.005 M MgCl_2_, 0.3 M sucrose, 0.005 M uridine 5′diphosphoglucose disodium (UDP) and water. The determination of Susy enzyme activity was the same as that for the invertases.

Antioxidant activity: The ability of the extracts to scavenge the 1,1-diphenyl-2-picrylhydrazyl radical (DPPH) was determined according to Pérez-Tortosa et al. [35]. Briefly, 50 μL of a series of diluted thyme extracts were added to 1 mL of a 100 μM methanol solution of DPPH. An absorbance at 517 nm was measured after a 30-minute incubation period at room temperature in the dark, and the readings were compared. The absorbance readings were compared to a calibration curve constructed using caffeic acid (0–1500 µM). The results were expressed as micromoles of reduced DPPH per gram fresh weight using an extinction coefficient of 12,500 M^−1^ cm^−1^ at 517 nm.

Nutrient contents in leaves: The mixture of leaves from different parts of the plants was collected and placed in a forced air oven at 65 °C until constant weight and then double ground in a mill, according to Tedesco et al. [36]. Approximately 200 mg were weighed on an analytical balance for subsequent sulfuric digestion of macronutrients and 500 mg for nitrous-perchloric digestion of micronutrients. From the digested material, the reading of nitrogen (N)—Kjeldhal method [37]; phosphorus (P)—spectrophotometer at 660 nm; potassium (K)—flame photometer B 462 (Micronal, São Paulo, Brazil); calcium (Ca); magnesium (Mg); zinc (Zn); copper (Cu); manganese (Mn) and iron (Fe)—Flame atomic absorption spectrophotometer Model AA 990F (PG Instruments Limited, Woodway lane, Alma park, Leicestershire, UK).

### 2.4. Yield Components

Yield components per pot: Ten replicates per treatment were used for the grain yield components, where each pot with a plant was considered a replicate. The weight of one thousand grains pot^−1^, number of panicle grains^−1^, and weight of grains pot^−1^ were determined.

Crude protein: This was determined using the Kjeldahl method [37] based on three steps: digestion, distillation, and titration. A 200 mg amount of grain flour was used in duplicate.

The calculation for the determination of total nitrogen was as follows:NT = (Va − Vb) × F × 0.1 × 0.014 × 100/P1

Being:

NT—Total nitrogen content in the sample, in percentage;

Va—Volume of hydrochloric acid solution used in sample titration;

Vb—Volume of hydrochloric acid solution used in blank titration;

F—Correction factor for hydrochloric acid;

P1—Sample mass (in grams).

To determine the total seed protein content, the value of total nitrogen verified by the Kjeldahl method [37] was multiplied by the factor conversion of nitrogen into protein; in this case, the value used was 6.75 (amaranth) and 6.25 (quinoa). The formula below was used to determine seed protein content: PT = NT x Fc where PT—total protein; NT—total nitrogen; Fc—conversion factor.

### 2.5. Experimental Design and Data Analyses

Two independent experiments (2019/2020 and 2020/2021) were carried out in a completely randomized design. Each pseudocereal species (amaranth or quinoa) was exposed to two treatments: (1) *a*[CO_2_]—400 ppm, and (2) *e*[CO_2_]—700 ppm, with ten individual plants to each treatment, totalizing 20 quinoa or 20 amaranth plants. In the current study, the ten pots planted with amaranth or quinoa in each OTC were treated as biological replicates (thus, *n =* 10 for *a*[CO_2_] and *n* = 10 for *e*[CO_2_]. The data obtained were tested for homoscedasticity using the Bartlett test and for normality using the Shapiro–Wilk test. The analysis of variance (ANOVA) was carried out using the statistical software R (www.r-project.org/). All data were analyzed for statistical differences between the *a*[CO_2_] and the *e*[CO_2_] conditions. Afterward, if F was significant, the means were compared to the control by the *t*-test (*p* ≤ 0.05). The data were expressed as the mean ± standard error (SE) of 10 replicates.

## 3. Results

### 3.1. Growth Parameters

Results presented in Figure 1 demonstrate a significant difference among treatments (*p* ≤ 0.05) for all the growth parameters. When cultivated under *e*[CO_2_], plants grew more, with longer shoots (Figure 1A), longer roots (Figure 1C), and higher shoot (Figure 1B) and root (1D) dry weight (Figure 1) when compared to plants grown under *a*[CO_2_]. When comparing the two species, the quinoa plants showed higher values, especially for the SDW and RDW parameters.

Data analysis showed significant differences between treatments when comparing the values obtained for the growth parameters (Figure 2A–D). Amaranth plants showed higher values in all parameters when subjected to *e*[CO_2_]; the same occurred for quinoa plants, except for panicle length (Figure 2A) and number of branches per panicle (Figure 2B), showing higher values in *a*[CO_2_].

### 3.2. Photosynthetic Pigments

In general, photosynthetic pigments (Figure 3) were not affected by CO_2_ treatments. Significant changes were observed only for chlorophyll-*a* (Figure 3A), chlorophyll-*b* (Figure 3B), and carotenoids (Figure 3C) in amaranth plants in the agricultural year 2020/2021.

### 3.3. Leaf Gas Exchange

Overall, *e*[CO_2_] produced changes in the leaf gas exchange (Figure 4), demonstrating an expressive difference between the treatments (*p* ≤ 0.05) to all measured parameters. CO_2_ assimilation (Figure 4A) was higher for both amaranth and quinoa at *e*[CO_2_]. High CO_2_ increases the net carbon assimilation rate, which is also evidenced by Ci (Figure 4C) for both crops. Therefore, a decrease in stomatal conductance and transpiration rate was observed under *e*[CO_2_]. Thus, the A/E ratio was higher at *e*[CO_2_]. Water use efficiency (Figure 4E) also showed a significant difference between the two treatments for both species, higher at *e*[CO_2_]. However, the results for quinoa and amaranth point to the same trend regarding water use efficiency and its influence on the photosynthesis rate.

### 3.4. Sucrose Metabolism-Related Enzyme Activity

Analysis of enzymatic activity related to sucrose metabolism (Figure 5) showed a significant increase in neutral invertase activity (Figure 5A), except for amaranth in the first agricultural year, in quinoa and amaranth plants grown under *e*[CO_2_]. Acidic invertases such as soluble acid invertases of the vacuole (Figure 5B) and cell wall acid invertase (Figure 5C) showed similar trends with higher values under *e*[CO_2_], except for amaranth plants in 2020/2021, when no difference was observed. Sucrose synthase activity (Figure 5D) was higher in *e*[CO_2_] for both species in agricultural years.

### 3.5. Carbohydrate Metabolism

Significant changes (*p* ≤ 0.05) were observed in carbohydrate metabolism (Figure 6). The total content of soluble sugars (Figure 6A) and sucrose (Figure 6B) was higher for both cultures under *e*[CO_2_], while the entire starch content (Figure 6C) showed a unique trend for each culture. Amaranth plants had a decrease in these parameters under *e*[CO_2_], while quinoa plants increased, although the total soluble amino acid content (Figure 6D) was higher in both cultures under *a*[CO_2_].

### 3.6. Antioxidant Activity

The antioxidant capacity evaluated by the DPPH scavenging assay (Figure 7) showed that amaranth and quinoa under *e*[CO_2_] significantly increased antioxidant capacity for both crops. Data obtained for amaranth and quinoa under *e*[CO_2_] corresponded to 8.12 (±0.67) and 9.10 (±1.04) µg caffeic acid eq g^−1^ DW, respectively, indicating the deactivation of free radicals.

### 3.7. Nutrient Contents in Leaves

Overall, *e*[CO_2_] decreased some macronutrients (Table 1) and micronutrients (Table 2) in amaranth and quinoa leaves. Here, we had different results for crops and agricultural years. Amaranth plants in *e*[CO_2_] revealed a decline in leaf contents of N; K; Ca; Zn; Mn and Cu, whereas quinoa plants presented a decrease in N; P (quinoa first year); K; Ca (quinoa second year); Zn; and Mn in *e*[CO_2_] (amaranth and quinoa two agricultural years). In general, magnesium leaf contents were non-significant. Fe and P levels were higher in *e*[CO_2_] for both crops and agricultural years.

### 3.8. Yield Components

Regarding the yield components, under *e*[CO_2_], there was a reduction in the 1000 grain weight for both crops (Table 3). However, there was a different performance regarding the number of grains per panicle. While the values for amaranth were non-significative, *e*[CO_2_] decreased the number of grains per panicle in quinoa plants. The grain yield per pot was reduced for quinoa under *e*[CO_2_], with 50.34% and 61.72% reduction in the first and second growing seasons, respectively.

### 3.9. Crude Protein

For the grain protein content (SPC), it was possible to observe that *e*[CO_2_] reduced this component (Table 4). The protein levels in grains were reduced in both species under *e*[CO_2_]. On average, during two growing seasons, protein content was reduced by 32% in amaranth and 50.5% in quinoa seeds.

## 4. Discussion

The hypothesis that C4 plants can respond to atmospheric CO_2_ concentrations as much as C3 plants has proven useful, though not always entirely predictive. This variability is influenced by genetic improvements [38], which result in differing responses depending on genotype, variety, and cultivar [39]. Consequently, the long-held assumption, deeply embedded in models of past vegetation–climate interactions, may require a fresh perspective [40].

Plant growth is a complex process influenced by numerous factors, including increased CO_2_ levels. This is what makes growth parameters important since they are related to greater light absorption [41,42]. Here, all of those results were increased by *e*[CO_2_]. Both an increase in shoot and root length (consequently, shoot and root dry matter) are major traits that can be attributed to the initial effects of *e*[CO_2_] in plants. Similar results were presented by Song et al. [42] for amaranth cultivated in 500 ppm and Bunce [43] for quinoa in 600 ppm of CO_2_, respectively. These experiments also were conducted in OTC.

The root system not only takes up soil nutrients and water for sustainable plant production but also pumps photosynthetically fixed C to soil organic matter (SOM) pools. It plays a crucial role in terrestrial C cycling. The *e*[CO_2_] exerts a substantial impact on these systems by influencing the morphology of root length and distribution. In addition, secondary roots control ecosystem C and N cycling as plants obtain water and nutrients and release exudates. In this study, it was possible to observe that the root system of amaranth and quinoa (Figure 8) plants from both species collected after grain maturation presented more secondary roots when in *e*[CO_2_], possibly as a strategy to maintain the nutritional and water status of the plants.

An increase in root elongation and branching was evidenced in *Sedum alfredii*, *e*[CO_2_] [44]. However, other studies found that a variety of plant species show just an increased fine root production [45,46]. Corroborating the results presented here, a meta-analysis carried out by Nie et al. [47] showed that the fine root biomass of plants had a significantly stronger response to *e*[CO_2_] in OTC experiments (+35.8%).

According to Ferreira et al. [48], the stem diameter is important because it is related to the falling of the plants. To be maintained upright and support it, the *e*[CO_2_] significantly increased the stem diameter of amaranth and quinoa. Similar changes in plant morphology have been reported elsewhere, for example, *e*[CO_2_] significantly increased the shoot of C4 plants like maize [49], sugarcane [50], foxtail millet [51] and C3 plants like quinoa [52] coffee tree [53], *Stylosanthes capitata* Vogel. [54]. While to the panicle length, quinoa plants had a controversial result. The control treatment presented a higher PL than *e*[CO_2_]. This phenomenon has been reported by some authors [55,56] as a downregulation or acclimatization of species to the increase in CO_2_ during the stadiums_._ Physiological processes often develop mechanisms of compensation that reduce or minimize the long-term effects of CO_2_; sometimes, a limitation of its regeneration capacity can be observed [57]. 

Chlorophylls and carotenoids are pigments capable of absorbing visible radiation and triggering photochemical reactions of photosynthesis [58]. In general, plants grown in *e*[CO_2_] show alteration in photosynthetic pigments. Some authors have mentioned the reduction of chlorophyll and carotenoids in leaves due to the increase in biomass under *e*[CO_2_], called the N dilution effect [59,60]. In some cases, changes in the C:N ratio cause an effect of nitrate (NO_3_) assimilation inhibition by the roots [61,62,63]. However, it was possible to observe two important factors: (1) change in pigments for amaranth plants (2020/2021), and (2) non-significant change but mostly higher levels in the control treatment, evidencing the importance of chlorophyll breakdown. This a highly coordinated and integral process of the plant development stages that are programmed to facilitate the dynamic remobilization of nutrients from organs/tissues to parts of the plant that are still growing, in particular to reproductive/storage organs during the transition of plants to reproductive growth [64,65].

Growth, photosynthetic pigments, and leaf gas exchange are closely linked. C4 plant species possess a “distinct pathway” of photosynthesis from C3 species, which in low atmospheric CO_2_ concentration could be concentrated to enable a more efficient carboxylation reaction due to changes in the ratio of CO_2_:O_2_ [66]. This does not mean that C4 species do not change the Calvin–Benson cycle’s fundamental machinery; rather, they have functionalized structural and biochemical additions around C3 photosynthesis to improve its efficiency. It is necessary to consider that most C4 plants fix CO_2_ in mesophyll cells with phosphoenolpyruvate carboxylase (PEPC), an enzyme that, unlike Rubisco, is insensitive to O_2_. Subsequently, CO_2_ is released in the bundle sheath cells where Rubisco is localized, and the Calvin–Benson cycle occurs. This additional step increases the availability of CO_2_ around Rubisco and minimizes its chance of catalyzing the oxygenation reaction [67].

Concentration of carbon dioxide is an important regulator in the dynamics of opening and closing of stomata. Through these mechanisms, plants exchange gas with the external environment [68]. Opening the stomata allows the diffusion of CO_2_ for photosynthesis, in addition to providing a path for water to diffuse from the leaves to the atmosphere [69]. Therefore, plants regulate the level of stomatal opening (stomatal conductance), seeking to maintain high rates of photosynthesis and reduce water loss. In the current study, *e*[CO_2_] increased CO_2_ assimilation (Figure 6A) and Ci (Figure 6C) in quinoa plants. Partial closure of the stomata could have been responsible for reducing stomatal conductance and transpiration rate. Thus, it was possible to maintain high rates of photosynthesis without compromising the internal concentration of CO_2_ since the greater difference in CO_2_ concentration between the atmosphere and the interior of the leaf compensates for the increase in stomatal resistance [70].

This compensation is relatively less understood for C4 species [71]. Theoretically, when it comes to C4 species in general, researchers consider C4 plants saturated at *a*[CO_2_]. Also, they might not be stimulated by *e*[CO_2_] [55]. We found significant stimulation rates, like Zhang et al. [71] for Broomcorn millet; Li et al. [51] for Foxtail millet; Davis and Ainsworth [72] for *Amaranthus rudis*, even though Santos [73] and Leakey et al. [74] had opposite results for *Amaranthus viridis* and maize, respectively.

C4 plants such as amaranth have increased vein densities during the evolution process, causing a reduction in intercellular air spaces and enhancement of bundle sheath organelles [75,76]. In addition, the increasing vein densities may not only increase structural integrity or enhance leaf water status [76] but may also allow xylem-transported CO_2_ to be utilized for photosynthesis by reducing the distance between vascular bundles transporting xylem-transported CO_2_ and photosynthetic cells [77].

The results of changes in C4 photosynthesis have not yet reached a consensus among researchers. Some authors mention the fact that increases observed in A at *e*[CO_2_], such as our results, are noticeable during the transition stadium between vegetative and flowering or apparent only during the early stage in some crops because of the carbohydrate metabolism. At this stage, plants are preparing to export macro and micronutrients to produce grains/seeds [78,79]. In addition, studies related to gene regulation have increasingly reported the particularity of genotypes, showing different responses to *e*[CO_2_] for the same species, mainly for C4 plants [80].

Carbon assimilated in photosynthesis is stored through carbohydrates, which are compounds generated in high quantities by plants and have high proportions of carbon [81]. In general, increases in carbohydrate production resulting from the increment in photosynthesis by *e*[CO_2_] can result in alterations in the production and partition of carbohydrates, as it raises the activity of enzymes that hydrolyze sucrose into sink organs [82,83].

Among the sugars synthesized in a plant, only a few of them are transported in the phloem over a long distance [84]. Upon arriving at sink tissues, sucrose can follow different pathways that can modulate sink strength and carbon flux [85]. Sucrose might be unloaded from the phloem to the apoplast by transporters or be hydrolyzed by invertases (CINV, CWINV, and VINV) to yield glucose and fructose, which can enter the sink cells via hexose transporters [86]. Besides that, as a reversible cleavage, SuSy might catalyze sucrose using UDP to yield fructose and UDP-G [87] and utilize other nucleotide phosphates for the cleavage, especially ADP, but usually with a lower affinity [88].

In the present study, the activity of sucrolytic enzymes was altered in leaves of amaranth and quinoa in the transition stage between vegetative and flowering by *e*[CO_2_]. At this stage, considering that grain yield is dependent on the plant source/sink relationship, the top two leaves are the primary source, and the florets are the primary sink for photosynthesis. This might also change plant carbon and nitrogen metabolism [64,89]. In addition to this direct effect on photosynthesis, many physiological processes are indirectly regulated, mainly through sugar detection and signaling pathways. Sugar signaling plays an important role in the plant’s response to *e*[CO_2_]; however, this is not well understood concerning the plant’s nutritional quality [90].

Accumulation of soluble sugars is a direct effect of *e*[CO_2_] due to the growth of triose phosphate synthesis in leaves, which can be further transformed into other carbohydrates, e.g., glucose, fructose, and sucrose. A meta-analysis made with publications between 1990 and 2018 showed that *e*[CO_2_] increased the concentrations of sucrose by 3.7% and total soluble sugar by 17.5% in leaves. This data shows a non-integration of newly fixed carbohydrates to growth, accumulating them in the leaves [91] and confirms the results presented here. Dong et al. [92] still proposed that the synthesized carbohydrates in leaves cannot be fully translocated to fruits as well as to roots, although one needs to be cautious regarding species variation.

Regarding sucrose, it is important to remember its relationship to plant growth and development. If photosynthetic capacity exceeds demand, excess photoassimilates remain in the chloroplast and can be stored in the form of starch. Thus, it is believed that the sink can control the activity of the source. Furthermore, there is an intricate relationship between source and sink, as both activities are controlled by environmental factors such as CO_2_ [85,93].

As we have seen so far, *e*[CO_2_] plays a crucial role in the physiology of plants. The plant photosynthesis, stomatal aperture, biomass production, yield, and water use efficiency could be modulated by the CO_2_ environment [94]. As a result, more carbohydrates could be transferred into the grains due to the increased photosynthesis in plants grown under a CO_2_-enriched environment. Nevertheless, there is generally a reduction in mineral contents, particularly N concentrations (Table 3), in grown plants at *e*[CO_2_], probably due to restricted root nutrient uptake caused by reduced mass flow and dilution effect, which is reflected directly on the total soluble amino acids content in leaves [51].

Re-translocation of nutrients from leaves could be a strategy to retain P efficiently. However, this is more observed in senescence [95,96,97]. Re-translocation by resorption serves to withdraw nutrients from leaves before abscission for later redeployment in developing tissues. The extent to which P is re-translocated and re-used depends on the plant’s nutrient status. Generally, with decreasing nutrient availability in an ecosystem, the amount of resorption of both N and P tends to increase [98,99]. This ratio could also be an indicator of which nutrient is most limiting in the ecosystem [100]. Considering that both nitrogen and phosphorus are involved in the photosynthetic machinery, most studies have reported that these nutrients also decrease due to increased carbon assimilation in several crops [101]. However, in this experiment, amaranth and quinoa (2020/2021) plants demonstrated an increase in P (Table 3).

Given that N and P availability control the global carbon cycle response to environmental changes, aspects like stoichiometric balance are important to consider. Nevertheless, the importance of P dynamics under *e*[CO_2_] is less clear. The assumption of a homeostatic N:P ratio in elevated CO_2_ due to similar proportional N and P responses need to be clarified more [102,103]. Xu et al. [104] also reinforce the idea that C3 and C4 plants also have different nutritional changes under *e*[CO_2_]. In a study where nitrogen availability was analyzed, Sudderth et al. [105] observed that N availability in the presence of *e*[CO_2_] increased foliar N content in *Amaranthus viridis* (C4 plant).

Several studies reported that *e*[CO_2_] decreased the mineral concentration by a dilution effect, suggesting that the decrease in mineral concentration is not specifically regulated by certain metabolic processes but by a dilution effect due to the increased biomass, as it was demonstrated earlier. It is well elucidated that the higher growth rate under high CO_2_ also increases the activity of anabolic processes that require nutrients, including osmoregulation (K), cell elongation and nucleic acid metabolism (B), metabolic pathways that require nutrients as cofactors (Ca, Mg, and Mn) and redox reactions (Fe, Zn, and Cu). Furthermore, to maintain homeostasis, plants can alter the absorption of these nutrients [104]. However, Table 4 shows that iron content increased significantly in *e*[CO_2_]. So far, we do not have a specific explanation for that. Perhaps more studies about the nutritional theme are necessary, especially when we talk about food security and food supply.

During the grain formation, the panicles act as nitrogen sinks in the plant. The process of accumulating storage during grain development in the middle and late maturation stages is crucial to determine crude protein [106]. These reserves are essential for human nutrition, with a better balance of essential amino acids than most cereals and legumes seed germination and seedling establishment [107,108,109]. Despite that, when plants grow under *e*[CO_2_], photosynthetic machinery is faster, causing a reduction in the nitrogen concentration and protein content of the grains [110]. Our result shows us a decrease in CSP; however, the mechanism of how every plant responds to the crude protein content is not totally clarified.

In theory, increased photosynthesis increases the availability of carbohydrates, which results in a gain in biomass and consequently in grain yield [5,55,111]. However, for the yield components presented, we must also consider the increase in average temperature (Table 4) in both agricultural years. Numerous studies have reported that the weight of 1000 grains and crude protein is reduced in *e*[CO_2_] when compared to *a*[CO_2_] conditions. These are fundamental characteristics when it comes to selecting species for a future climate change scenario. In this study, it was possible to observe that amaranth plants showed a smaller decrease in productivity when compared to quinoa plants [71,72].

## 5. Conclusions

Despite their different biochemical pathways (C3 in quinoa and C4 in amaranth), projected rise in atmospheric [CO_2_] significantly increases photosynthesis and growth in both crops. Such changes are associated with various metabolic processes, ultimately affecting plant nutritional quality, carbohydrate production, antioxidant capacity, and grain yield.

Elevated [CO_2_] alters the metabolism of amaranth and quinoa plants, as evidenced by the changes in growth patterns and photosynthetic pigments that reflect changes in gas exchange, carbohydrate metabolism, and nutritional status. As a result, plants may experience a reduction on essential nutrients like nitrogen. However, a notable increase in secondary metabolites was observed in both crops, and together with the enhanced enzymatic antioxidant system, these compounds may help protect the plants against stress.

For quinoa plants, the *e*[CO_2_] led to reduction in grain yield and protein content. In contrast, there was no reduction in grain yield for amaranth, but there was a considerable reduction in grain protein content. This reduction is particularly significant in these crops renowned for its high protein content and quality. 

Future research should focus on evaluating the protein quality and other nutritional aspects in these pseudocereals under *e*[CO_2_]. Our findings ultimately underscore the critical need to breed plants that can adapt to *e*[CO_2_] as means to mitigate its negative effects and to ensure sustainable and nutritious crop production in future environmental conditions.

## Figures and Tables

**Figure 1 plants-13-03453-f001:**
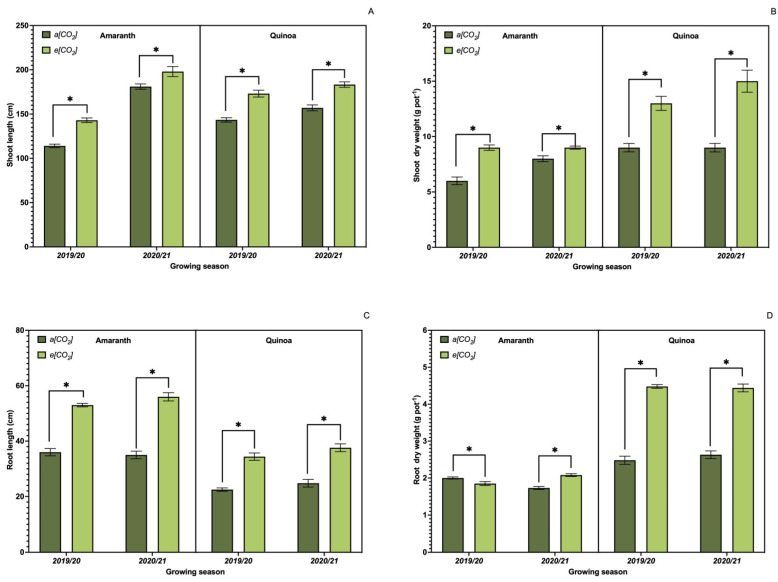
Effect of CO_2_ on plant growth parameters. Shoot length (**A**); shoot dry matter (**B**); root length (**C**); root system dry matter (**D**) of amaranth and quinoa plants. *a*[CO_2_] = plants grown in OTC with 400 ± 50 μmol mol^−1^ CO_2_; *e*[CO_2_] = plants grown in OTC with 700 ± 50 μmol mol^−1^ CO_2_. The experiment was conducted during the 2019/2020 and 2020/2021 growing seasons. Error bars correspond to the 95% confidence interval of the mean. * Indicates significant difference between CO_2_ concentration (*t* test *p* ≤ 0.05, n = 10).

**Figure 2 plants-13-03453-f002:**
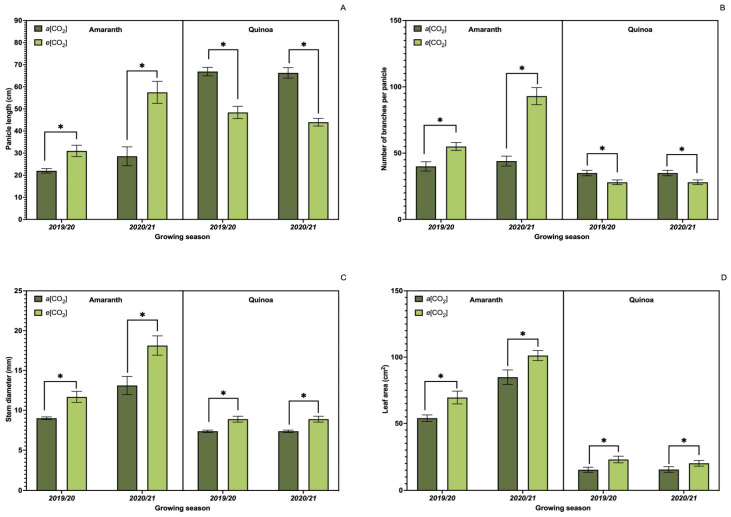
Effect of CO_2_ on growth parameters. Panicle length (**A**); number of branches per panicle (**B**); stem diameter (**C**); leaf area (**D**) of amaranth and quinoa plants. *a*[CO_2_] = plants grown in OTC with 400 ± 50 μmol mol^−1^ CO_2_; *e*[CO_2_] = plants grown in OTC with 700 ± 50 μmol mol^−1^ CO_2_. The experiment was conducted during the 2019/2020 and 2020/2021 growing seasons. Error bars correspond to the 95% confidence interval. * Indicates significant difference between CO_2_ concentration (*t* test *p* ≤ 0.05, n = 10).

**Figure 3 plants-13-03453-f003:**
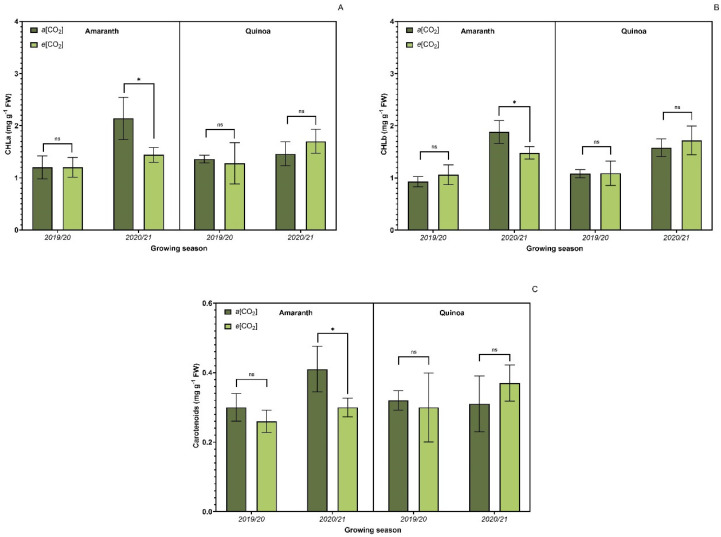
Effect of CO_2_ concentration on photosynthetic pigments. Chlorophyll-*a* (CHLa) (**A**); chlorophyll-*b* (CHLb) (**B**); carotenoids (**C**) of amaranth and quinoa plants in transition stadium between vegetative and flowering. *a*[CO_2_] = plants grown in OTC with 400 ± 50 μmol mol^−1^ CO_2_; *e*[CO_2_] = plants grown in OTC with 700 ± 50 μmol mol^−1^ CO_2_. The experiment was conducted during the 2019/2020 and 2020/2021 growing seasons. Error bars correspond to the 95% confidence interval. * Indicates significant difference between CO_2_ concentration (*t* test *p* ≤ 0.05, n = 10). ns indicates non-significant differences (*t* test *p* ≤ 0.05, n = 10).

**Figure 4 plants-13-03453-f004:**
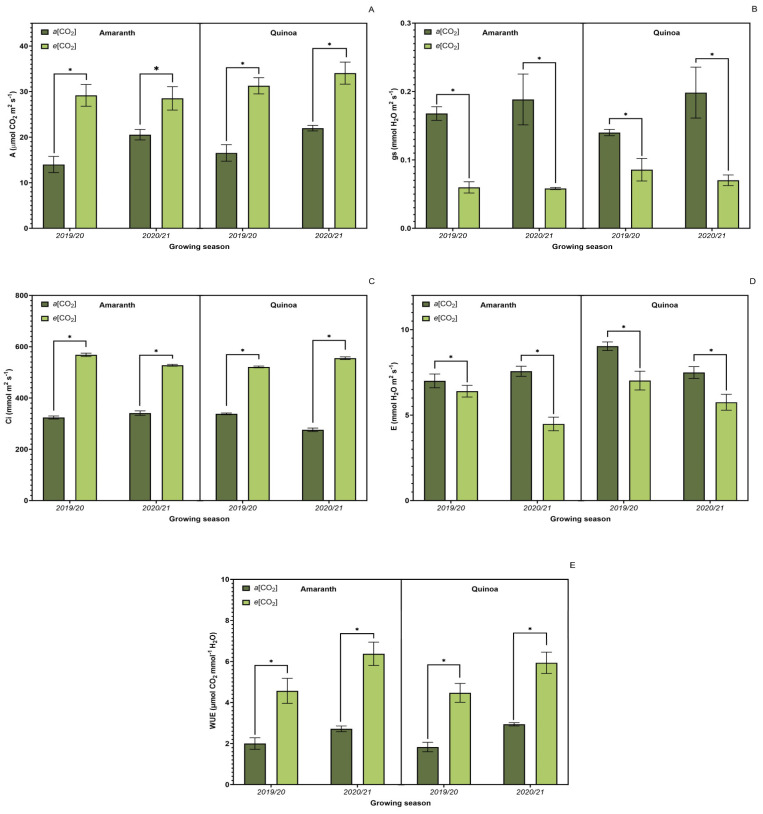
Effect of CO_2_ on leaf gas exchange. Net CO_2_ assimilation (**A**); stomatal conductance (**B**); internal concentration of CO_2_ (**C**); transpiration rate (**D**); water use efficiency (WUE) (**E**) of amaranth and quinoa plants in transition stadium between vegetative and flowering. *a*[CO_2_] = plants grown in OTC with 400 ± 50 μmol mol^−1^ CO_2_; *e*[CO_2_] = plants grown in OTC with 700 ± 50 μmol mol^−1^ CO_2_. The experiment was conducted during the 2019/2020 and 2020/2021 growing seasons. Error bars correspond to the 95% confidence interval. * Indicates significant difference between CO_2_ concentration (*t* test *p* ≤ 0.05, n = 10).

**Figure 5 plants-13-03453-f005:**
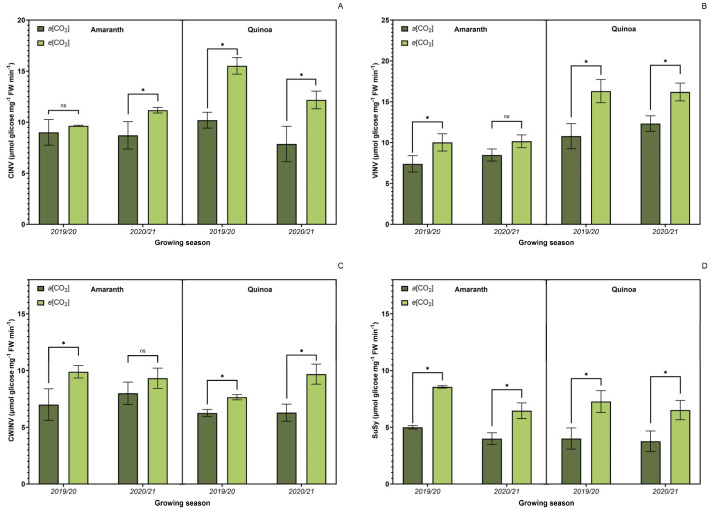
Effect of CO_2_ on sucrose metabolism-related enzyme activity in leaves: soluble neutral invertases of cytosol (CINV) (**A**); soluble acid invertases of the vacuole (VINV) (**B**); cell wall acid invertase (CWINV) (**C**); and sucrose synthase (SuSy) (**D**) activity. *a*[CO_2_] = plants grown in OTC with 400 ± 50 μmol mol^−1^ CO_2_; *e*[CO_2_] = plants grown in OTC with 700 ± 50 μmol mol^−1^ CO_2_. The experiment was conducted during the 2019/2020 and 2020/2021 growing seasons. Error bars correspond to the 95% confidence interval. *Indicates significant difference between CO_2_ concentration (*t* test *p* ≤ 0.05, n = 4). ns indicates non-significant.

**Figure 6 plants-13-03453-f006:**
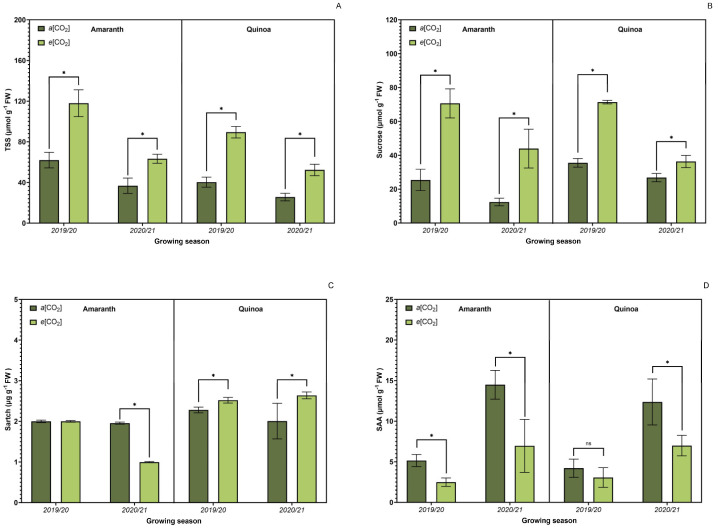
Effect of CO_2_ on carbohydrate metabolism: total content of soluble sugars (**A**), sucrose-SUC (**B**), starch (**C**), and total soluble amino acids (**D**) on amaranth and quinoa leaves in transition stadium between vegetative and flowering. *a*[CO_2_] = plants grown in OTC with 400 ± 50 μmol mol^−1^ CO_2_; *e*[CO_2_] = plants grown in OTC with 700 ± 50 μmol mol^−1^ CO_2_. The experiment was conducted during the 2019/2020 and 2020/2021 growing seasons. Error bars correspond to the 95% confidence interval. * Indicates significant difference between CO_2_ concentration (*t* test *p* ≤ 0.05, n = 4). ns indicates non-significant.

**Figure 7 plants-13-03453-f007:**
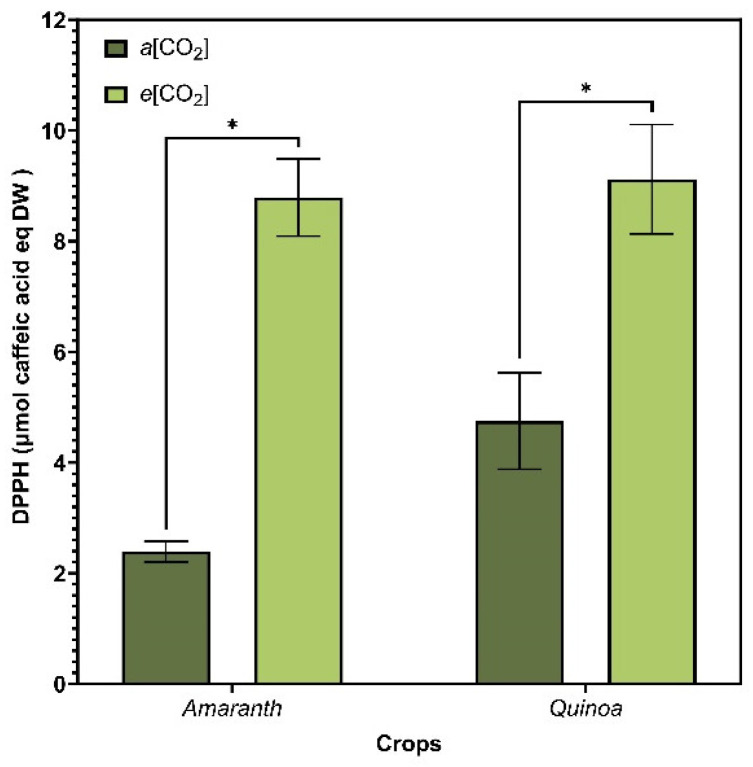
Antioxidant capacity evaluated by DPPH from amaranth and quinoa in transition stadium between vegetative and flowering, grown in the presence of different CO_2_ concentrations (400 and 700 μmol mol^−1^ CO_2_). The extracts evaluated were obtained using the growing season 19/20 and 2020/2021. *a*[CO_2_] = plants grown in OTC with 400 ± 50 μmol mol^−1^ CO_2_; *e*[CO_2_] = plants grown in OTC with 700 ± 50 μmol mol^−1^ CO_2_. The experiment was conducted during the 2019/2020 and 2020/2021 growing seasons. Error bars correspond to the 95% confidence interval. * Indicates significant difference between CO_2_ concentration (*t* test *p* ≤ 0.05, n = 10).

**Figure 8 plants-13-03453-f008:**
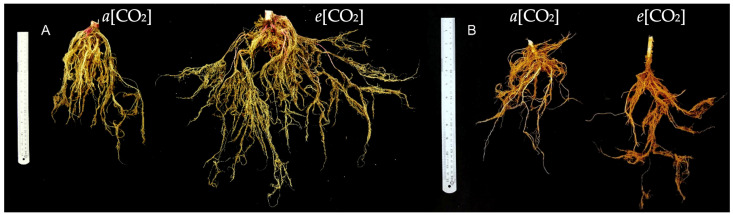
Effects of CO_2_ concentration on roots of amaranth (**A**) and quinoa (**B**) collected after grain maturation. *a*[CO_2_] = plants grown in OTC with 400 ± 50 μmol mol^−1^ CO_2_; *e*[CO_2_] = plants grown in OTC with 700 ± 50 μmol mol^−1^ CO_2_.

**Table 1 plants-13-03453-t001:** Macronutrient contents in leaves of amaranth and quinoa plants in transition stadium between vegetative and flowering as affected by CO_2_ concentration in two growing seasons.

Treatment [CO_2_]	Amaranth (First Year)	Amaranth (Second Year)	Quinoa (First Year)	Quinoa (Second Year)
		N (g kg^−1^)		
*a*[CO_2_]	52.0 ± (1.79) *	53.0 ± (1.66) *	54.9 ± (4.57) *	53.2 ± (0.54) *
*e*[CO_2_]	49.2 ± (0.98)	36.0 ± (1.79)	48.4 ± (0.27)	38.5 ± (1.71)
CV%	1.15	1.57	2.52	1.12
		P (g kg^−1^)		
*a*[CO_2_]	3.4 ± (0.13) *	3.1 ± (0.08) *	5.1 ± (0.42) *	2.7 ± (0.08) *
*e*[CO_2_]	4.3 ± (0.17)	3.7 ± (0.17)	4.0 ± (0.37)	3.0 ± (0.18)
CV%	1.65	1.56	3.54	2.03
		K (g kg^−1^)		
*a*[CO_2_]	46.5 ± (2.33) *	46.5 ± (3.87) *	50.8 ± (2.07) *	50.9 ± (1.97) *
*e*[CO_2_]	30.7 ± (4.20)	33.9 ± (1.62)	46.3 ± (1.86)	45.7 ± (0.64)
CV%	3.54	2.97	1.63	1.22
		Ca (g kg^−1^)		
*a*[CO_2_]	16.9 ± (1.88) *	27.3 ± (2.75) *	11.3 ± (0.46) ^ns^	21.7 ± (1.63) *
*e*[CO_2_]	12.9 ± (1.63)	16.9 ± (1.88)	11.2 ± (2.58)	15.3 ± (2.97)
CV%	4.74	4.29	6.66	5.23
		Mg (g kg^−1^)		
*a*[CO_2_]	8.1 ± (0.88) *	8.8 ± (1.07) ^ns^	6.3 ± (1.13) ^ns^	8.9 ± (0.78) ^ns^
*e*[CO_2_]	9.9 ± (0.21)	7.8 ± (1.34)	6.0 ± (1.03)	8.8 ± (0.71)
CV%	2.99	5.87	7.06	3.38

* Indicates significant difference by *t*-test (*p* ≤ 0.05, n = 4). ns indicates non-significant. CV: coefficient of variation. Values in parentheses correspond to the 95% confidence interval. *a*[CO_2_] = plants grown in OTC with 400 ± 50 μmol mol^−1^ CO_2_; *e*[CO_2_] = plants grown in OTC with 700 ± 50 μmol mol^−1^ CO_2_.

**Table 2 plants-13-03453-t002:** Micronutrient contents in leaves of amaranth and quinoa plants in transition stadium between vegetative and flowering in response to CO_2_ concentration in two growing seasons.

Treatment [CO_2_]	Amaranth (First Year)	Amaranth (Second Year)	Quinoa (First Year)	Quinoa (Second Year)
		Fe (mg kg^−1^)		
*a*[CO_2_]	164.4 ± (4.50) *	174.5 ± (1.33) *	80.1 ± (7.83) *	180.0 ± (3.63) *
*e*[CO_2_]	185.4 ± (2.37)	207.8 ± (3.60)	135.1 ± (8.64)	235.5 ± (2.66)
CV%	0.89	2.04	3.09	
		Zn (mg kg^−1^)		
*a*[CO_2_]	22.2 ± (2.10) *	36.1 ± (2.99) *	11.3 ± (2.31) *	49.0 ± (1.03) *
*e*[CO_2_]	15.5 ± (4.45)	26.0 ± (2.92)	6.4 ± (4.55)	20.8 ± (0.88)
CV%	7.41	3.82	16.37	1.11
		Mn (mg kg^−1^)		
*a*[CO_2_]	554.7 ± (9.19) *	254.2 ± (4.18) *	348.4(5.42) *	755.0 ± (8.93) *
*e*[CO_2_]	424.8 ± (2.60)	193.7 ± (8.29)	284.0 ± (3.86)	585.1 ± (9.43)
CV%	0.56	1.18	0.6	
		Cu (mg kg^−1^)		
*a*[CO_2_]	17.6 ± (1.72) *	19.6 ± (1.72) *	12.4 ± (1.72) *	14.0 ± (1.72) *
*e*[CO_2_]	15.2 ± (1.72)	17.6 ± (1.72)	31.4 ± (1.72)	29.2 ± (1.72)
CV%	4.21	3.73	3.15	3.2

* Indicates significant difference by *t*-test (*p* ≤ 0.05, n = 4). CV: coefficient of variation. Values in parentheses correspond to the 95% confidence interval. n = 4. *a*[CO_2_] = plants grown in OTC with 400 ± 50 μmol mol^−1^ CO_2_; *e*[CO_2_] = plants grown in OTC with 700 ± 50 μmol mol^−1^ CO_2_.

**Table 3 plants-13-03453-t003:** Yield components in grains of amaranth and quinoa plants in transition stadium between vegetative and flowering in response to CO_2_ concentration.

Treatment [CO_2_]	Amaranth (First Year)	Amaranth (Second Year)	Quinoa (First Year)	Quinoa (Second Year)
	**Weight of 1000 grains (g)**			
*a*[CO_2_]	1.11 ± (0.01) *	1.10 ± (0.00) *	2.41 ± (0.07) *	2.48 ± (0.01) *
*e*[CO_2_]	0.86 ± (0.03)	0.81 ± (0.01)	1.71 ± (0.04)	1.70 ± (0.00)
CV%	1.87	0.74	1.89	0.28
	**Number of grains per panicle**			
*a*[CO_2_]	10.921 ± (11.28) ^ns^	9.879 ± (7.5) ^ns^	2.598 ± (2.66) *	2.772 ± (18.45) *
*e*[CO_2_]	9.876 ± (10.52)	10.922 ± (9.14) ^ns^	2.174 ± (3.48)	1.830 ± (14.0)
CV%	13.12	11.44	18.2	9.99
	**Grain yield per pot (g)**			
*a*[CO_2_]	9.37 ± (0.96) ^ns^	8.40 ± (0.66) ^ns^	5.72 ± (0.69) *	5.67 ± (0.78) *
*e*[CO_2_]	9.30 ± (0.73)	8.41 ± (0.70)	2.84 ± (0.62)	2.17 ± (0.47)
CV%	12.83	11.38	21.58	23.08

* Indicates significant difference by *t* test (*p* ≤ 0.05, n = 4); ns indicates non-significant. CV: coefficient of variation. Values in parentheses correspond to the 95% confidence interval (n = 10). *a*[CO_2_] = plants grown in OTC with 400 ± 50 μmol mol^−1^ CO_2_; *e*[CO_2_] = plants grown in OTC with 700 ± 50 μmol mol^−1^ CO_2_.

**Table 4 plants-13-03453-t004:** Effect of CO_2_ concentration during the plant growing season on seed/grain crude protein content of amaranth and quinoa plants.

Treatment [CO_2_]	Crude Protein (%)			
	Amaranth (First Year)	Amaranth (Second Year)	Quinoa (First Year)	Quinoa (Second Year)
*a*[CO_2_]	18.28 ± (1.35) *	17.99 ± (0.37) *	15.97 ± (0.97) *	15.94 ± (0.92) *
*e*[CO_2_]	12.74 ± (0.62)	11.87 ± (0.30)	7.97 ± (0.89)	7.79 ± (0.41)
CV%	4.29	1.45	4.92	3.81

* Indicates significant difference by *t* test (*p* ≤ 0.05, n = 4); CV: coefficient of variation. Values in parentheses correspond to standard deviation. *a*[CO_2_] = plants grown in OTC with 400 ± 50 μmol mol^−1^ CO_2_; *e*[CO_2_] = plants grown in OTC with 700 ± 50 μmol mol^−1^ CO_2_.

## Data Availability

The original contributions presented in this study are included in the article/Appendix A. Further inquiries can be directed to the corresponding author.

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
