# Peer review of "Physiological and Biochemical Responses of Pseudocereals with C3 and C4 Photosynthetic Metabolism in an Environment with Elevated CO2"

_plants, 2024, doi:10.3390/plants13233453_

Round 1

Reviewer 1 Report

Comments and Suggestions for Authors

This is certainly a worthwhile topic, as these important crops have been underutilized in elevated CO2 studies.  However, some experimental clarifications are needed for this to become acceptable, and some statements in the introduction are simply wrong.

Line 42:  global warming did not cause the increase in CO2, just the reverse!

Line 53:  attributing the yield increase with CO2 primarily to reduced stomatal conductance increasing WUE is simply incorrect.

Methods:  Why are there "coolers" in the OTC?  How were they controlled?  How was CO2 at night controlled?  How many OTC were used? 10 of each CO2, or were there multiple pots of each species in each chamber?  The experimental design is unclear.  We need to know details of the weather during the experiment, as well as the temperature and humidity inside the chambers.   No information is given about the leaf temperature or chamber water vapor pressure during the gas exchange measurements.  The huge difference between CO2 treatments in gs values but relatively much smaller differences in transpiration rate require explanation.  

Author Response

  • Regarding Reviewer´1 comments

  • This is certainly a worthwhile topic, as these important crops have been underutilized in elevated CO2 However, some experimental clarifications are needed for this to become acceptable, and some statements in the introduction are simply wrong.

Authors: We agree and thank the reviewer for their valuable comments, suggestions, and corrections on this manuscript.

  • Line 42: global warming did not cause the increase in CO2, just the reverse!

Authors: We apologize for this error. Line 42-44 has been removed.

  • Line 53: attributing the yield increase with CO2 primarily to reduced stomatal conductance increasing WUE is simply incorrect.

Authors: We apologize for this error. Line 53-55 has been removed, as well as references.

  • Methods: Why are there "coolers" in the OTC?  How were they controlled?  How was CO2 at night controlled?  How many OTC were used? 10 of each CO2, or were there multiple pots of each species in each chamber? 

Authors: The coolers are responsible for homogenizing the air inside, preventing fluctuations in the amount of carbon dioxide injected into each of the OTCs. These coolers, as well as the CO2 control, were controlled by a device created by the Herbology Center of the Federal University of Pelotas, which is a leader in climate research in Brazil. Regarding OTC number, a total of 4 OTCs were used, 2 of 700 ppm, and 2 of 400 ppm with 10 plants each.

  • The experimental design is unclear.

Authors: Experimental design was added in the manuscript text and can be tracked in the manuscript by the yellow highlights (Lines 296-299).

  • We need to know details of the weather during the experiment, as well as the temperature and humidity inside the chambers.

Authors: We apologize for this lack information. Two supplementary figures were add  to supply its query.

  • No information is given about the leaf temperature or chamber water vapor pressure during the gas exchange measurements. The huge difference between CO2 treatments in gs values but relatively much smaller differences in transpiration rate require explanation. 

Authors: The temperature and VPD leaf, respectively ranged from 26-27°C (Leaf VPD: ±1.76) in 2019/20 and 26-29°C (Leaf VPD: ±1.89) in 2020/21.

Finally, we thank you for the suggestions and if you have any new contributions, please send us and we will take care of everything you need.

Reviewer 2 Report

Comments and Suggestions for Authors

The manuscript entitled “Physiological and Biochemical Responses of Pseudocereals With C3 and C4 Photosynthetic Metabolism in an Environment With Elevated CO2” depicted the impacts of increased CO2 concentration on the growth, productivity, grain quality, and biochemical changes of quinoa and amaranth plants. The work is interesting and has a certain contribution to the field. The experiment is well-studied.  My suggestion is a minor revision.

It is suggested to add some precise data from your results in the Abstract section.

Line 16: It should be “..the high CO2 concentration e[CO2]”.

Line 35: This sentence is confusing. What changes occur for 2 million years?

The Introduction section seems a little long and I suggested shortening the length if possible.

Line 146: Why did you choose this CO2 concentration for your experiment.

Line 154: Please add the reasons why you choose these cultivars for study.

Line 307: it should be “among” not “between”

The Discussion seems a little long and I suggested shortening the length if possible.

Author Response

  • Regarding Reviewer´2 comments

  • The manuscript entitled “Physiological and Biochemical Responses of Pseudocereals With C3 and C4 Photosynthetic Metabolism in an Environment With Elevated CO2” depicted the impacts of increased CO2 concentration on the growth, productivity, grain quality, and biochemical changes of quinoa and amaranth plants. The work is interesting and has a certain contribution to the field. The experiment is well-studied. My suggestion is a minor revision.

Authors:

  • It is suggested to add some precise data from your results in the Abstract section.

Authors: We add a new sentence in the “Abstract” section (Lines 29-31).

  • Line 16: It should be “..the high CO2 concentration e[CO2]”.

Authors: Done.

  • Line 35: This sentence is confusing. What changes occur for 2 million years?

Authors: The sentence has been thoroughly revised and rewritten and can be tracked in the manuscript by the yellow highlights (Lines 34-36).

  • The Introduction section seems a little long and I suggested shortening the length if possible.

Authors: We appreciate your consideration, however, we consider that the introduction contains relevant pieces of information.

  • Line 146: Why did you choose this CO2 concentration for your experiment.

Authors: The choice of CO2 concentrations for the experiment was based on extensive research into other climate-related studies involving different species in Brazil and around the world. Until the beginning of this work, according to the IPCC's future forecasts of 700 ppm, this was the rate considered most likely in a future scenario for agriculture. Although some authors extrapolate to 1000 ppm, we set out to make our research as close to reality as possible.

  • Line 154: Please add the reasons why you choose these cultivars for study.

Authors: Seeds of the cultivar BRS Alegria (amaranth) and BRS Piabiru (quinoa). In Brazil, both cultivars are commonly used in the so-called "safrinha" or second harvest period, and present satisfactory biomass and grain production. These two characteristics, associated with the short period from emergence to maturity, make them a potential component of the no-tillage system. For this purpose, were sown in polystyrene trays on commercial substrate (Plantmax®). After the appearance of the sec-ond pair of true leaves, the seedlings were transplanted into 8-L polyethylene pots filled with soil, which was previously analyzed for its physical and chemical attributes, amended and fertilized according to technical recommendations (EMBRAPA, 1999), keeping only one plant per pot after the complete establishment of the plants. In the transi-tion from the vegetative to the reproductive stage, which occurred in the second half of December in both years of cultivation, leaves were collected for subsequent growth, physi-ological and biochemical analyses, as described below. Both experiments were conducted in a completely randomized design with 10 replications.

  • Line 307: it should be “among” not “between”

Authors: Done.

  • The Discussion seems a little long and I suggested shortening the length if possible.

Authors: We appreciate your consideration, but, discussion was elaborated with based nos mains findings of research.

 We greatly appreciate the constructive comments, as well as the observations made by the reviewer.  Once more, we deeply appreciate the efforts of the reviewers aimed to improve the formal and scientific merits of the manuscript. We hope that the revised manuscript can be accepted for publication in Plants.

Round 2

Reviewer 1 Report

Comments and Suggestions for Authors

I am still confused by the experimental design, and your comments about the changes made:

You stated that 4 chambers were used, but it not clear that 4 were used each year, or two each year.  That matters a great deal to the statistics.  In any case, it is not statistically correct to use each pot as a replicate for statistical analysis.  At least you have two years of each CO2 treatment that could be used as replicates (n=2), or maybe there were two replicates each year?

You said that temperatures inside the chambers were recorded, but have presented no data, so we have no idea what the temperatures were in this experiment. 

For the leaf gas exchange, I still see no indication of the leaf temperature, CO2 level, light level or VPD during the measurements, nor any information about the date(s?) of the measurements.  

Author Response

Dear Assistant Editor, Nina He and Special Issue Editor, Elliot Cao

Please find enclosed the 2 revised version of the manuscript ([Plants] Manuscript ID: plants-3320824) entitled “Physiological and Biochemical Responses of Pseudocereals With C3 and C4 Photosynthetic Metabolism in an Environment With Elevated CO2 which we would like to publish in your Journal. Substantial revisions have been performed in order to follow the ‘reviewers 1' comments that can be tracked in the manuscript by the blue highlights. Specific descriptions and justification about comments follow below.

Regarding Reviewer´1 comments

I am still confused by the experimental design, and your comments about the changes made: You stated that 4 chambers were used, but it not clear that 4 were used each year, or two each year.  That matters a great deal to the statistics.  In any case, it is not statistically correct to use each pot as a replicate for statistical analysis.  At least you have two years of each CO2 treatment that could be used as replicates (n=2), or maybe there were two replicates each year?

Authors: Yes, four chambers were used per year. We apologize for this. Therefore, a brief description of the experimental setup is in supplementary figure 1. In the material and methods section, subtopics were reworded and another was created “Elevated-CO2 treatment” (Lines 159-169). Besides, the experimental design was added with more details (Lines 306-317).

You said that temperatures inside the chambers were recorded, but have presented no data, so we have no idea what the temperatures were in this experiment.

 Authors: The temperature for each day of the experiment 2019/2020 and 2020/2021 is shown in supplementary figure 1a-b. Besides, % relative humidity, as well as minimum/maximum was also added (Lines 166-169).

 For the leaf gas exchange, I still see no indication of the leaf temperature, CO2 level, light level or VPD during the measurements, nor any information about the date(s?) of the measurements.

The leaf temperature ranged 26 to 27°C in 2019/20 and 26-29 °C in 2020/21. The CO2 level in the chamber was matched for each treatment (400 and 700 μmol mol-1 CO2), and the photon flux density was regulated to 1500 µmol of photons m-2 s-1 with a light source attached to the measuring chamber. Regarding VPD leaf, 1.76 mean, and 1.89 mean in 2019/2020 and 2020/21, respectively. Finally, the leaf gas exchange measurements were performed only once for each experiment.

We greatly appreciate the its constructive comments, as well as, the new observations. Once more, we deeply appreciate the efforts of the currently reviewer aimed to improve the formal and scientific merits of the manuscript.

We hope that the 2 revised version can be accepted for publication in Plants.

Yours faithfully,

Dr. Sidnei Deuner

Round 3

Reviewer 1 Report

Comments and Suggestions for Authors

Thanks for the revisions.  I am now much clearer about the experimental details.  However, I do not think it is valid to use pots within a single chamber as replicates.  In this case, the only real statistical replication is the two years of data, so n = 2 in all cases.  I do not think that this would change any conclusions.  I did not see a description of plant stage of development for the times of leaf gas exchange measurement. 

Author Response

Reviewer´1 comments

 Thanks for the revisions.  I am now much clearer about the experimental details. 

However, I do not think it is valid to use pots within a single chamber as replicates.  In this case, the only real statistical replication is the two years of data, so n = 2 in all cases.  I do not think that this would change any conclusions. 

Authors: We sincerely thank their valuable comments, suggestions, and corrections.

Regarding OTCs and plants number (n=)…

There were 2 chambers for each CO2 condition

Each chamber had 5 amaranth plants and 5 quinoa plants, totaling 10 plants per chamber.

2 OTCs x 2 CO2 conditions (400 and 700 μmol mol-1 CO2),  = 4

1 OTC = 5 amaranths and 5 quinoa = 10

 I did not see a description of plant stage of development for the times of leaf gas exchange measurement.

Authors: We apologize for this. Information follows and also was added in the manuscript (see lines 208-209; 373-374). The leaf gas exchange analysis was performed in a transition period, i.e, vegetative to flowering.

We hope that the 3 revised version can be accepted for publication in Plants.

Yours faithfully,

Dr. Sidnei Deuner
